# *Neisseria mucosa* Does Not Inhibit the Growth of *Neisseria gonorrhoeae*

**Saïd Abdellati** [1,†] , **Jolein Laumen** [1,2,†] , **Natalia Gonzalez** [1] , **Sheeba S. Manoharan-Basil** [1] ,
**Christophe Van Dijck** [1,2] , **Irith De Baetselier** [1] , **Delphine Martiny** [3] , **Tessa de Block** [1] and **Chris Kenyon** [1,4,*]

1 HIV/STI Unit, Institute of Tropical Medicine, 2000 Antwerp, Belgium; sabdellati@itg.be (S.A.);
jlaumen@itg.be (J.L.); ngonzalez@ext.itg.be (N.G.); sbasil@itg.be (S.S.M.-B.); cvandijck@itg.be (C.V.D.);
tdeblock@itg.be (T.d.B.); idebaetselier@itg.be (I.D.B.)
2 Laboratory of Medical Microbiology, Vaccine and Infectious Disease Institute, University of Antwerp,
2000 Antwerp, Belgium
3 Microbiology Department, Laboratoire Hospitalier Universitaire de Bruxelles, 1060 Brussels, Belgium;
delphine.martiny@lhub-ulb.be
4 Division of Infectious Diseases and HIV Medicine, University of Cape Town, Anzio Road,
Observatory 7700, South Africa
* Correspondence: ckenyon@itg.be
† These authors contributed equally to this work.

**Abstract:** Antibiotic-sparing treatments are required to prevent the further emergence of antimicrobial resistance in *Neisseria gonorrhoeae*. Commensal *Neisseria* species have previously been found to inhibit the growth of pathogenic *Neisseria* species. For example, a previous study found that three out of five historical isolates of *Neisseria mucosa* could inhibit the growth of *N. gonorrhoeae*. In this study, we used agar overlay assays to assess if 24 circulating and historical isolates of *Neisseria mucosa* could inhibit the growth of 28 circulating and historical isolates of *N. gonorrhoeae*. Although pitting around each colony of *N. mucosa* created an optical illusion of decreased growth of *N. gonorrhoeae*, we found no evidence of inhibition (n = 24). In contrast, positive controls of *Streptococcus pneumoniae* and *Escherichia coli* demonstrated a strong inhibitory effect against the growth of *N. gonorrhoeae*.

**Keywords:** *Neisseria mucosa*; *Neisseria gonorrhoeae*; agar overlay assay; bacterial competition

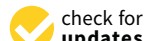



## 1. Introduction

A number of countries worldwide are reporting an increasing incidence of sexually transmitted infections due to *Neisseria gonorrhoeae* [1]. This, combined with increasing antimicrobial resistance in this organism, has led to efforts to find novel therapies to treat and prevent this infection [2–5]. One of these strategies has been to use antiseptics to prevent the acquisition and transmission of *N. gonorrhoeae* to and from the oropharynx [2,6]. The prevalence of *N. gonorrhoeae* in the pharynx may reach 10% in high-risk populations and *N. gonorrhoeae* has been shown to be highly susceptible in vitro to antiseptics such as those based on essential oils [6–9]. A pilot clinical study found that an essential oil-based mouthwash reduced the prevalence of pharyngeal *N. gonorrhoeae*, as assessed by culture [10]. These findings provided the motivation for two randomized controlled trials that assessed if essential oil-based mouthwashes could reduce the incidence of *N. gonorrhoeae* and other STIs in men who have sex with men [2,6].

One of these was the preventing resistance in gonorrhoea (PReGo) study conducted in our centre [2]. This placebo-controlled trial randomized high-risk men who have sex with men to intensive use of an essential oil-based mouthwash and gargle, or placebo, to try to reduce the incidence of bacterial STIs in this population. The study found that mouthwash increased rather than decreased the incidence of oropharyngeal *N. gonorrhoeae*. The same essential oil-based mouthwash had a similar though statistically non-significant effect in the other study that used a slightly different study design (the OMEGA study) [6].

One of the possible explanations for these surprising results is that essential oil-based mouthwashes could reduce the abundance of commensal bacteria that have an inhibitory effect on *N. gonorrhoeae*. One such commensal bacteria is *Neisseria mucosa*, which has recently been shown to inhibit the growth of *N. gonorrhoeae* by Aho et al. [11].

*N. mucosa* is a healthy core component of the oropharyngeal microbiome and even low concentrations of an essential oil-based mouthwash have been shown to be bactericidal to *Neisseria* spp. [7]. If the use of the essential oil-based mouthwashes reduce the prevalence/abundance of *N. mucosa*, and *N. mucosa* inhibits the growth of *N. gonorrhoeae*, then the essential oil-based mouthwash could increase the susceptibility for *N. gonorrhoeae* infection [2]. In a similar vein, a randomized controlled trial established that nasal inoculation with *N. lactamica* reduced the incidence of colonization with *N. meningitidis* [12]. If the in-vitro anti-gonococcal effect of *N. mucosa* could be confirmed, *N. mucosa* might be evaluated as a probiotic to prevent gonococcal infection.

This provided the motivation for the current study where we tested the hypothesis that *N. mucosa* could inhibit the growth of *N. gonorrhoeae*. Our central objective was to assess if our locally circulating isolates of *N. mucosa* and other commensal *Neisseria*, including those circulating in the PReGo participants, were able to inhibit the growth of *N. gonorrhoeae*.

## 2. Results

Agar overlay assays were used to assess if 21 circulating and historical isolates of *Neisseria mucosa*, and 16 isolates from other *Neisseria* species, could inhibit the growth of 26 circulating and historical isolates of *N. gonorrhoeae*.

None of the commensal *Neisseria* or *N. meningitidis* exhibited any activity against *N. gonorrhoeae* (Table S1). The isolate of *S. pneumoniae* demonstrated clear evidence of inhibition against all nine strains of *N. gonorrhoeae* (median diameter of inhibition = 21 mm) (Figure 1a,b; Table S1). The inhibitory effect of *E. coli* was less pronounced (Figure 1a,b; Table S1). Inhibition was evident in three out of nine *N. gonorrhoeae* strains tested—the median diameter of inhibition was 11 mm (Table S1).

A proportion of the colonies of *N. mucosa* exhibited a repellant effect, whereby they repelled the layer of agar poured over them (Figure 1c,d; Table S1). This created 'pitting' colonies or a convex slope between the top of the second layer of agar and the edge of each *N. mucosa* colony, which created an illusion of reduced *N. gonorrhoeae* growth around each *N. mucosa* colony [13] (Figure 2). Closer visual inspection, however, confirmed that *N. gonorrhoeae* growth over this convex slope around the *N. mucosa* colonies was not macroscopically distinguishable from that elsewhere (Figure 1c,d).

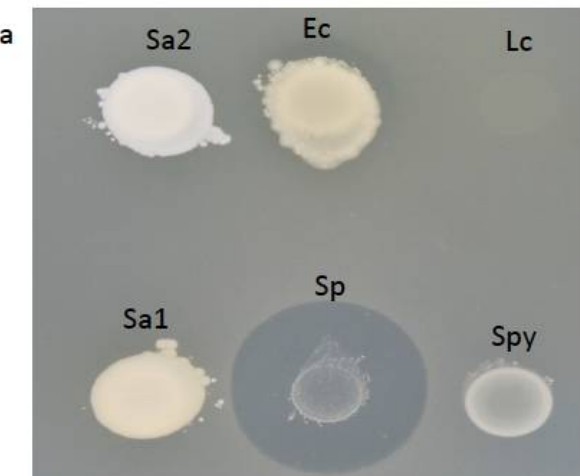
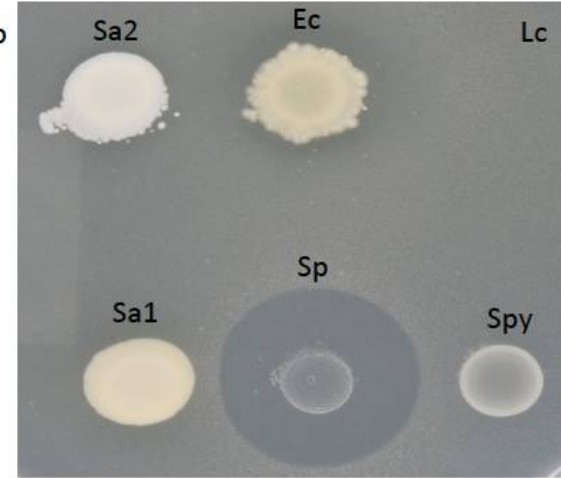

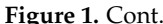

**Figure 1.** Cont.

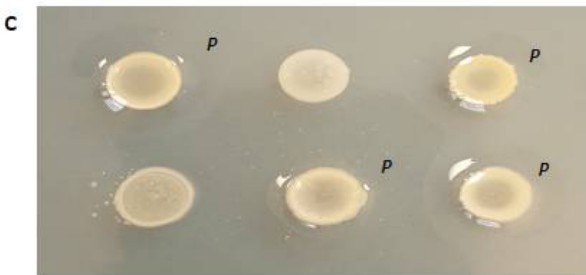
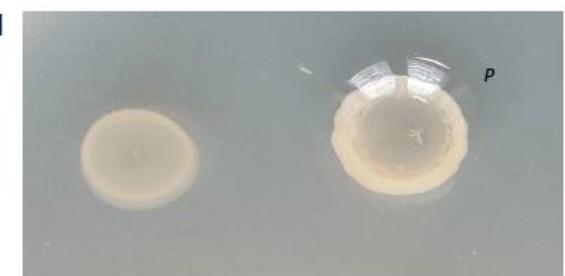

**Figure 1.** Agar overlay assay testing the ability of various bacterial species to inhibit the growth of a lawn of *N. gonorrhoeae* strain RL1 (**a**) and strain 21.189 (**b**). Only the colonies of *Escherichia coli* (Ec) and *Streptococcus pneumoniae* (Sp) inhibited the growth. The colonies of *N. mucosa* in (**c**,**d**) did not exhibit any inhibitory effect on the growth of *N. gonorrhoeae* strain 21.163 (**c**) and strain WHO-W (**d**). A close-up of one of the *N. mucosa* colonies in (**d**) demonstrates the pitting (*p*) of the upper layer of agar around the right-hand colony of *N. mucosa*. Ec—*Escherichia coli* (ATCC 25922); Lc—*Lactobacillus crispatus* (LMG 9479); p—pitting; Sa—*Staphylococcus aureus* (1:ATCC 29213, 2:ATCC 25913); Sp—*Streptococcus pneumoniae* (ATCC 49619); Spy—*Streptococcus pyogenes* (LMG 14238).

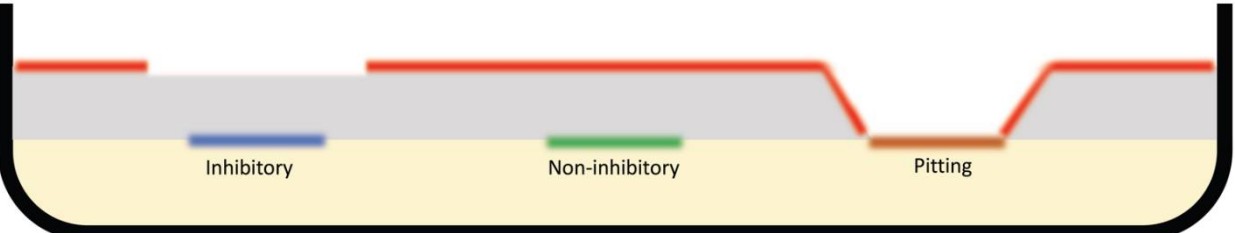

**Figure 2.** A schematic illustration of the difference between growth-inhibition and pitting in the agar overlay assay. An agar plate in cross section is depicted, on which 3 bacterial colonies (labeled 'inhibitory', 'non-inhibitory' and 'pitting') were spotted and incubated for 24 h before a layer of GCB agar containing $10^6$ CFU/mL of *N. gonorrhoeae* was poured over the plate (grey layer). Whilst the 'non-inhibitory' colony had no effect on the growth of *N. gonorrhoeae* (red line), and the 'inhibitory' colony had a clear inhibitory effect, the major effect of the 'pitting' colony was to repel the second layer of agar, thus creating an area around it which appears more translucent from above. Close visual inspection, including from the lateral aspect, of the depressed sections of the second layer of agar around the 'pitting' colony revealed the uninhibited growth of *N. gonorrhoeae*.

## 3. Discussion

Unlike Aho et al., we could find no evidence that *N. mucosa* or any other commensal *Neisseria* was able to inhibit the growth of *N. gonorrhoeae* [11]. This was despite using a large number of clinical and reference strains of *N. gonorrhoeae* as target strains, and the largest collection of commensal *Neisseria* tested to date as inhibitory bacteria.

How can these discordant findings be explained? Aho found this inhibitory effect in three out of five *N. mucosa* isolates. The isolates were all obtained from ATCC collections and did not include any recent clinical isolates. No photos were provided of the agar overlay assays showing that *N. mucosa* inhibited the growth of *N. gonorrhoeae*. However, one image of *N. mucosa* inhibiting the growth of *N. flavescens* was provided.

In our study, we followed an identical agar overlay protocol utilizing a larger panel of isolates of *N. mucosa* and *N. gonorrhoeae*. The experiments were performed by a laboratory technician with over 25 years of experience culturing *Neisseria* species (SA). The plates were examined by this person and two others with extensive experience in culturing *Neisseria* species (CK and JL). All three concurred that pitting around each colony of *N. mucosa* created an optical illusion of decreased growth around the colony. Close visual inspection confirmed that there was no inhibition of growth. We did not ascertain what the molecular determinants of this repellant effect were, as this was not an objective of this study.

We consider this a parsimonious explanation for the different findings between the two studies. It could be possible that only certain strains of *N. mucosa* are able to inhibit specific strains of *N. gonorrhoeae*, and that we did not include any of these combinations in our experiments. We did, however, test one of the three isolates of *N. mucosa* shown to have an inhibitory effect by Aho et al. This isolate (ATCC 25996) had no effect on the growth of 23 contemporarily circulating strains of *N. gonorrhoeae* in our laboratory. We did not have access to, and therefore did not include any of the same strains of *N. gonorrhoeae* used by Aho et al. As a result, we cannot exclude the possibility that our *N. mucosa* strains would have had an inhibitory effect on the *N. gonorrhoeae* strains used by Aho et al. Furthermore, our experiments were not conducted in duplicate. In pilot studies, we found that *N. mucosae* did not inhibit the growth of *N. gonorrhoeae*, and our experiment was thus designed to maximize the chances of detecting any inhibitory effect on *N. gonorrhoeae*. We thus evaluated if any strains of *N. mucosa* we could access (n = 24) could inhibit the growth of a large panel of circulating and type strains of *N. gonorrhoeae* (n = 28). This constitutes the largest experiment to have assessed this effect. The previous largest experiment was conducted with four isolates of *N. mucosa* tested against one isolate of *N. gonorrhoeae*, and a further one isolate of *N. mucosa* tested against seven isolates of *N. gonorrhoeae* [11]. Because we found no evidence of inhibition in any of the pair-wise comparisons in our experiments, we consider it unlikely that repeating the experiments in triplicate would change our findings.

We mainly included pharyngeal *N. gonorrhoeae* target strains. Since these were isolated from asymptomatic individuals, they may have adapted to live with oral commensals. Therefore, a greater number of strains from anatomical sites other than the pharynx should be included in future studies. We also cannot completely exclude the possibility that an unevaluated different experimental condition, such as storage of the isolates or the source of the agar used, was responsible for the differences in the results between the two studies. It could be argued that a further weakness of the study is that inhibition was only assessed via visual inspection. This is, however, the standard method of assessing growth inhibition in the agar overlay assay [11]. Our study, unlike that of Aho et al., did include positive controls. These showed clear and consistent evidence of inhibition. Taken together, these findings suggest that *N. mucosa* is unlikely to have a significant inhibitory effect on the growth of *N. gonorrhoeae*—at least in the agar overlay assays evaluated here. More importantly for our current research, we consider it unlikely that a broad range of *N. mucosa* isolates contained a sufficiently potent compound against our currently circulating strains of *N. gonorrhoeae* to be able to explain the findings of the PReGo and OMEGA studies. A further strength of this study is that we assessed if currently circulating isolates of commensal *Neisseria* were able to inhibit the growth of *N. gonorrhoeae*. The previous largest study by Aho et al. was limited to testing historical isolates from collections [11].

More recent studies by Kim et al. have found that *Neisseria elongata* is able to kill *N. gonorrhoeae* in vivo and in a mouse infection model [14,15]. The toxic compound was found to be differentially methylated DNA that was taken up by *N. gonorrhoeae* via transformation [15]. *N. elongata* is one of the less prevalent commensal *Neisseria* spp. In the PReGo and ComCom studies, for example, we only isolated *N. elongata* from one individual [16]. This isolate showed no evidence of inhibiting the growth of *N. gonorrhoeae* in the agar-overlay assay (Table S1). We cannot exclude the possibility that this isolate could exhibit an inhibitory effect on the growth of *N. gonorrhoeae* if assessed in more sensitive assays, such as those used by Kim et al. [15]. Commensal *Neisseria* may also inhibit the growth of *N. gonorrhoeae* via type 6 secretory systems. In a series of elegant experiments, Custodia et al. have established that a type 6 secretory system, in certain strains of *N. cinerea*, is able to reduce the survival of the gonococcus by five-fold [17,18]. Another recent study has found that *N. cinerea* forms microcolonies of epithelial cells in a way that impairs the colonization of the epithelium by *N. meningitidis* [19]. These effects can only be detected in experiments using cell models, which we did not do. Other studies of meningococcal colonization have illustrated how complex the interactions between pathogenic *Neisseria* and other bacterial

species may be. Audry et al., for example, have established that the initial meningococcal colonization of the nasopharynx may not result in inflammation in the short term, as the bacteria may be trapped within the mucus lining [20]. During this entrapped state, the interaction with *Streptococcus mitis* but not *Moraxella catarralis*, triggered invasive disease via degradation of the encasing mucins. These findings illustrate that whilst the results of in vitro assessments of growth inhibition are important, great caution should be exercised in extrapolating these findings to what happens in vivo [21].

In conclusion, we concur with Aho et al., that commensal microbes represent a possible source of antimicrobial compounds that could play an important role in reducing the emergence of AMR in *N. gonorrhoeae* and other bacteria [11,22–24]. Based on our findings, we consider it more likely that such anti-gonococcal compounds will be discovered from organisms such as *S. pneumoniae*, rather than *N. mucosa* [23,25–32].

## 4. Materials and Methods

### 4.1. Origin of Bacterial Isolates

#### 4.1.1. Inhibitory/Producer Bacterial Isolates

Most Neisseria isolates were obtained from two clinical studies conducted at our centre:

(i)   The Preventing Resistance in Gonorrhoea Study (PReGo), a single-centre randomized controlled trial conducted at the Institute of Tropical Medicine in Antwerp, Belgium, between 2019 and 2020, that assessed the efficacy of an antiseptic mouthwash to prevent STIs among 343 MSM using PrEP [2].

(ii)  The Commensals in the Community Study (ComCom), a survey of the oropharyngeal microbiomes of Institute of Tropical Medicine (ITM) employees conducted in June 2020 [16]. In both studies, oropharyngeal swabs (ESwab$^{TM}$ COPAN Diagnostics Inc., Brescia, Italy) were taken and inoculated onto blood and modified Thayer–Martin agar plates using the streak plate technique, and incubated at 35–37 °C and 5% $CO_2$. Plates were examined after 48 h, and Neisseria-like colonies were selected based on a positive oxidase test and a Gram stain. Neisseria-like colonies were enriched on blood agar plates and stored in skim milk at −80 °C. Cultures of Neisseria-like colonies were shipped to Laboratoire des Hôpitaux Universitaires de Bruxelles-Universitair Laboratorium Brussel (LHUB-ULB), where species were identified using matrix-assisted laser desorption/ionization time-of-flight mass Spectrometry (MALDI-TOF MS), on a MALDI Biotyper® Sirius IVD system using the MBT Compass IVD software and library (Bruker Daltonics, Bremen, Germany) consisting of 9607 spectra.

All *N. mucosa* isolates obtained from the PReGo and ComCom studies (n = 14), as well as a random selection of *N. meningitidis* (n = 3) and other commensal *Neisseria* obtained from these two studies—*N. subflava* (n = 4), *N. cinerea* (n = 2), *N. lactamica* (n = 1), *N. oralis* (n = 3), *N. longate* (n = 1), and *Neisseria* spp. (n = 1; Table S1)—were included in the present work.

In addition, we also included 6 *N. mucosa* isolates from our ITM historical collection. Five of these were ATCC strains and one was a historical clinical specimen obtained from a patient in 1977, and the DSM4631/ATCC 25996 isolate used by Aho et al. was obtained from the DSMZ (https://www.dsmz.de/collection/catalogue/details/culture/DSM-46, accessed 2 October 2021).

#### 4.1.2. *N. gonorrhoeae* Target Strains

Three strains of *N. gonorrhoeae* were used as target strains for all experiments (WHO-F, WHO-X and MoNg003—a clinical isolate obtained from an individual with asymptomatic pharyngeal *N. gonorrhoeae* infection attending our STI clinic in 2020). In addition, one ATCC strain of *N. gonorrhoeae*, WHO-W and 23 other circulating strains of *N. gonorrhoeae* were tested against some of the putative inhibitory bacteria (Table S1).

#### 4.1.3. Non-Neisseria Isolates

Six ATCC strains of species previously shown to inhibit the growth of *N. gonorrhoeae* were included to serve as potential positive control for the agar overlay inhibition tests:

*Streptococcus pneumoniae* (n = 1; ATCC 49619), *Escherichia coli* (n = 1; ATCC 25922), *Staphylococcus aureus* (n = 2; ATCC 29213, ATCC 25913), *Streptococcus pyogenes* (n = 1; LMG 14238) and *Lactobacillus crispatus* (n = 1; LMG 9479) [23,25–32] (available at https://www.atcc.org accessed 1 November 2021).

*4.2. Agar Overlay Assay*

The details of the agar overlay assay have been described elsewhere [11]. Briefly, all strains used in the experiment were propagated on Columbian blood agar plates for 18–24 h. The cultures were suspended in 10 µL of phosphate-buffered saline (PBS) containing $10^9$ CFU/mL of inhibitory strains. These were spotted onto GC agar and incubated in 5% $CO_2$ at 35–37 °C for 24 h. Then, 10 mL of melted GCB agar containing $10^6$ CFU/mL of a target strain was added to each spotted plate. The plates were then re-incubated for 24 to 48 h. The diameter of the zone of inhibition surrounding each producer strain was assessed at 24 h.

**Supplementary Materials:** The following supporting information can be downloaded at: https://www.mdpi.com/article/10.3390/sci4010008/s1, Table S1: Inhibitory activity of various commensal *Neisseria* and other species in agar overlay assay; Table S2: Clinical study, anatomical site, year of isolation and antimicrobial susceptibilities of isolates used in the study.

**Author Contributions:** Conceptualization, C.K., S.A., S.S.M.-B., I.D.B. and J.L.; methodology, S.A., N.G.; software, C.K.; validation, S.A., J.L.; formal analysis, S.A., C.K.; investigation, S.A., N.G., D.M., T.d.B.; data curation, C.V.D.; writing—original draft preparation, C.K.; writing—review and editing, C.K.; visualization, S.A., C.K.; supervision, C.K.; All authors have read and agreed to the published version of the manuscript.

**Funding:** This research received no external funding.

**Institutional Review Board Statement:** Ethics approval was obtained from ITM's Institutional Review Board (1276/18 and 1351/20) and from the Ethics Committee of the University of Antwerp (19/06/058 and AB/ac/003).

**Informed Consent Statement:** Not applicable.

**Data Availability Statement:** All the relevant data generated during this study is provided in Table S1.

**Acknowledgments:** We would like to thank the PReGo and ComCom study participants for providing the samples used in this study.

**Conflicts of Interest:** The authors declare no conflict of interest.

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
