# Peer review of "Neisseria mucosa Does Not Inhibit the Growth of Neisseria gonorrhoeae"

_sci, doi:10.3390/sci4010008_

Round 1

Reviewer 1 Report

The manuscript is badly formatted. For example, it says that it is under evaluation in Pathogens, which is wrong.

Introduction

The hypothesis of the authors is not clear. Also, the objectives of the study are missing.

M & M

Please include in supplementary material a detailed table with characteristics of the test isolates, e.g., isolation details, clinical history, biochemical profile etc.

Results

Table 1 should be moved to supplementary material

There are evidence-based comparisons of the findings, this is a serious flaw.

Discussion

This is rather shallow and does not cover fully the findings. Also, some recent relevant references are missing. This part needs to be rewritten.

In general, I am cautious about this publication. The lack of statistical analysis is a significant flaw, which greatly reduces the value of the manuscript.

The manuscript can be publishable, but it requires significant effort, extensive revision and vast improvement before possible acceptance.

Author Response

Please see attached letter

Reviewer 2 Report

This manuscripts reports that Neisseria mucosa does not inhibit Neisseria gonorrhoeae.  It is counter to previous reports, and will spur additional investigation into this area of research.  Publication of these types of reports are important to advance fields.   

The introduction adequately reviews the field.  The experiments are scientifically sound.  The conclusions are supported by the results.  The authors provide a complete experimental section that will allow for the results to be repeated in other labs.  Overall, this is an important contribution that will advance the field of antibiotics and antimicrobials.  It will be of great interest to researchers working with N. gonorrhoeae, and methods of inhibition.  

Author Response

Please see attached letter

Reviewer 3 Report

  • A brief summary The authors examined if isolates of Neisseria mucosa are inhibiting the growth of N. gonorrhoeae so they can be used as antibiotic-sparing treatment in a form of probiotic in preventing infections with N. gonorrhoeae and further emergence of antimicrobial resistance in this pathogen. The authors found that isolates of Streptococcus pneumoniae and Escherichia coli but not N. mucosa have an inhibitory effect on N. gonorrhoeae growth. 
  • General concept comments

    The weakness and the complexity of the experiment are commented  in detail by authors in the Discussion section. However,  although the authors are claiming they included positive control in the experiment unlike Aho et al. in their study (line 170), this is not visible in the section Materials and Methods nor the section Results (S. pneumoniae and E. coli are not considered positive controls; if yes, please explain).

    Also, the list of references (there are two lists?) and the citation of the references in the text (for example page 2 - reference number 8 is followed by the reference number 15 and other reference numbers between are missing) is not adequate. 

  • Specific comments                                                                                line 41 - generic name for antiseptics should be used (instead of Listerine) - this refers to the whole Introduction section;                                                                                                                                        line 73, line 154 - "..if 24 circulating and historical isolates of Neisseria mucosa..." - according to Table 1 and Materials and Methods section there were total of 21 and not 24 N. mucosa isolates included in the experiment                                                                                                                                                                                                                line 75, line 156 - "...28 circulating and historical isolates of N. gonnorrhoeae." - according to Materials and Methods section there were 28 but in Table 1 there are only 26 isolates

Author Response

Please see attached letter

Round 2

Reviewer 1 Report

The manuscript has been improved with the revision.
Subject to adding some more recent relevant refences, which are missing, it can be accepted for publication.

Reviewer 3 Report

The authors answered and corrected accordingly all the comments except the comment regarding positive controls. They only answered that S. pneumoniae and E. coli were included as positive controls as these species have been previously been shown to inhibit the growth of N. gonorrhoeae in the agar-overlay assays. However,  positive controls should be described in Materials and Methods section (from where the isolates of S. pneumoniae and E. coli were obtained; also the reference(s) in which these species have been previously shown to inhbit the growth of N. gonorrhoeae in the agar-overlay assays should be cited).

Author Response

This manuscript is a resubmission of an earlier submission. The following is a list of the peer review reports and author responses from that submission.

Round 1

Reviewer 1 Report

In this study, Abdellati et al. applied various N. mucosa to agar overlay assay and showed no inhibition towards N. gonorrhoeae (GC). The results were claimed contradicted from what has been concluded by Aho et al. This study has its merit in the clinical settings of the inhibitory relationship between N.mucosa and N.gonorrhoeae. However, various N. mucosa strains may produce different results including the pitting colonies.

Major concern:

  1. The GC strains are different.

The GC strains used here are asymptomatic. Logically speaking they have adapted to live with other oral commensals, including N.mucosa. Therefore, no inhibition is expected. Aho et.al. has used the common GC strains but all of them were from the reproductive tract. It would be interesting to examine few of the strains from other anatomical locations to see if the inhibition is there.

  1. Figure 1.

Figure legend needs to be revised. I do not see difference between (a) and (b) where the legends indicates E.coli (a) and S. pneumoniae (B). Also, the author should indicate that p standing for pitting area.

6.

Authors conclude that “pitting colonies” could reason the conflict result between this study and previous research done by Aho et al. (line 50 to line 53 and line 60 to line 61). However, the authors did not mention which N. mucosa strain has ability to form pitting colonies. Is it including the strains Aho et al used?

Minor:

  1. Line 21 does not have a period.
  2. Line 45, 47, and 74 Bacteria name should be italicized in the context.

Reviewer 2 Report

Thank you for allowing me to review this interesting work; my comments are as follows:

ABSTRACT

  1. Should include introductory sentence(s) indicating why the study was performed.
  2. If your conclusion is that "pitting colonies" should not be mistaken for inhibition of growth, then this statement should be included in the abstract as well.

INTRODUCTION

  1. The first paragraph of this introduction [lines 25-41] is well laid out, including thoughtful questions, and allows the reader to follow your thought process.
  2. The second paragraph of the introduction states several findings, as well as methodological approaches, which traditionally are not presented in the introduction section. Consider moving these to the results and methods sections. [lines 43 - 53]
    • The only statement that belongs in an introduction is "The inhibitory effect...is well established" [lines 46-47]
  3. Would all readers be able to understand what an agar "repellant effect" entails? There are no references or further explanations of this effect in your manuscript.

RESULTS

  1. Your team details important findings here and justifies careful consideration of macroscopic evaluation of such colonies.
  2. Line 61 - thank you for providing a reference and the findings of "pitting colonies". Again, your findings are important to continue this type of research, however, it may be even better represented if you provided a lateral view diagram of how this may look on a plate.
  3. What does "closer inspection" [lines 61-62] entail? Did you visualize the colonies from the side? with microscope? with different lighting? Please provide more information.

TABLE 1

  1. This table is large and contains some information that is not detailed in the manuscript text (MICs, Year, Site). Consider removing this information so that the reader can focus on the data that is consistent with the topic of this manuscript. If this information is important to present, then please provide context as to why it is important in the manuscript text.

FIGURE 1

  1. There needs to be more labels. I believe Sa = S. aureus, Ec = E. coli, Spy = Strep pyogenes, Sp = step pneumo, Lc = local control? PLEASE CLARIFY and add a legend to this figure.
    • Which strain is Sa1 and Sa2? Please label or make reference to strains in Table 1.
  2. P = pitting colonies?
  3. It would be optimal to include rulers to the image so that you can demonstrate similar sizes of the colonies that are actually unaffected by the "optical illusion" of pitting.
  4. In addition to this figure, are you able to create a lateral or cross-sectional view of the colony to better demonstrate this "pitting" effect? The top-down or "birds-eye view" of the colonies may not be the best way to demonstrate the pitting effect.

METHODS

  1. For MALDI-TOF identification, which species library/(ies) were used to distinguish between species?

ACKNOWLEDGEMENTS

  1. As a courtesy, I would suggest written acknowledgment to the Aho et al group for sharing their N. mucosa isolate with your group.

Reviewer 3 Report

In the current paper, the Authors tried to re-evaluate the previously published data regarding the effect of N.mucosa on the growth of N. gonorrhoeae and observed contradicting results.  

Major comments

  • Authors need to provide more information in the Abstract and Introduction regarding the background, rather than directly starting with what authors did and observed.
  • The results section need to be significantly improved, it is hard to follow the content and the table and figure. For eg. in line 58-64, it is hard to correlate the explanation and figure 1. 
  • Table 1: Did the authors determined the MIC for each strain? If yes, please provide the relevant information and explanation in the Methods, results, and discussion section.  If they obtained the information from a source, then please add in a reference. 
  • The authors also need to explain the reason for including the MIC in table 1.
  • Figure legend needs to be below the figure
  • The figure is not properly labeled, and therefore hard to interpret. What is Sa2, Ec, Lc, sp, Spy, P.? What are top rows and bottom rows. 
  • Individual figures (1a,1b,1c,1d) need to be cited in the text.
  • In the discussion section, one of the reasons that the authors describe is the difference in personal experience. I don't feel this is a valid reason to compare and should not be discussed

Additional Experiment

  • Line 105-108: As mentioned, the authors didn't include the same strains of N.gonorrhoeae used by Aho et al. It would be great to get those strains from Aho et al. and redo the overlay assay to confirm the results.   

Reviewer 4 Report

General comment: The manuscript “Neisseria mucosa does not inhibit the growth of Neisseria gonorrhoeae” by Abdellati and co-workers investigated the inhibition of the growth of N. gonorrhoeae due to other Neisseria species or other bacteria found in the oropharyngeal area. The manuscript is not suitable for publication in Pathogens, because of the following points

Major comments:

  • Abstract is too short. It is not structured correctly as it only consists of one sentences about Material & Methods and one sentence Results.
  • Introduction is missing some major points, like how common is gonorrhoeae in the oropharynx?, what are the main issues of such an infection?, etc.
  • Currently the aim of the study is missing
  • The publication of Aho et al. is mentioned several times and the author has also compared some of their own data to the data of this study. Unfortunately the current study does not mention whether they performed their test in duplicates or triplicates on different days.
  • As the study does not show if their own data could be reproduced as mean value and standard deviation are missing.
  • The discussion is not well structured and highly relevant points like the overall bactericidal potential of the mouthwash is missing. 

Overall the study design does not seem well thought through, for example if Listerine® is inhibiting the growth of probiotic Neisseria species, it also may influence the growth of N. gonorrhoeae? Is there a reason why the author does not investigated the effect of this mouthwash against their Neisseria strains to show the potential of this product at first? The author did also not mention any study, which may have investigated the bactericidal effect as well as the mechanisms of action and the limitation of this product or its major components.

Round 2

Reviewer 1 Report

All comments were properly addressed.

Reviewer 3 Report

The authors tried to improve the paper but the main result of the paper i.e. the overlay agar is still not convincing. The authors still failed to mark properly figure 1c and 1d and it is still confusing. Since the data interpretation is via visual inspection which is not clear on the figure and to the readers, they need to prove this using different techniques which is more convincing. Also, the authors point that the experiment was done once which is not sufficient as it needs to be repeated at least three times. 

Reviewer 4 Report

I want to thank the authors for their adaptations. Now, the whole manuscript is easier to read and the understanding of the main issue, the results and the drawn conclusions are clearer.

  1. Minor comments:
  • At the moment the transitions of the sentences in the abstract still sound a bit ticked off. I would therefore make the abstracts transitions "softer" and more pleasant to read in order to make the article more palatable to future readers.
  • Based on the blurred lines and the blurred labeling of figure 2 the resolution of this figure 2 has to be improved.
  • I also suggest redesigning Table 1 as there are some outer lines missing.